# Improving Human Diets and Welfare through Using Herbivore-Based Foods: 2. Environmental Consequences and Mitigations

**DOI:** 10.3390/ani14091353

**Published:** 2024-04-30

**Authors:** John R. Caradus, David F. Chapman, Jacqueline S. Rowarth

**Affiliations:** 1Grasslanz Technology Ltd., PB 11008, Palmerston North 4442, New Zealand; 2DairyNZ, Lincoln 7647, New Zealand; david.chapman@dairynz.co.nz; 3Faculty of Agriculture and Life Science, Lincoln University, 85084 Ellesmere Junction Road, Lincoln 7647, New Zealand; jsrowarth@gmail.com

**Keywords:** animal-based food, environment, greenhouse gas emission, land use, life cycle analysis, plant-based food, nutrition, soil carbon, water

## Abstract

**Simple Summary:**

Optimal human health requires the adequate provision of all nutrients in the correct proportions, ensuring the provision of energy and essential small molecules. All primates, including humans, are omnivorous but the most striking difference from other primates is the remarkable diversity of the diets we consume. Animal-sourced foods are important for human nutrition and health, but they can have a negative impact on the environment. The aim here is to examine these impacts that can result in land use tensions associated with population growth and the loss of native forests and wetlands during agricultural expansion, increased greenhouse gas emissions, and high water use with poor water quality outcomes. However, several technologies and practices can be used to mitigate against these impacts. These include grazing when feed quality is high, the use of dietary additives, breeding of animals with higher growth rates and increased fecundity, rumen microbial manipulations using vaccines and other additives, soil management to reduce nitrous oxide emission, management systems to improve carbon sequestration, improved nutrient use efficacy, use of cover crops, low-emission composting barns, covered manure storages, and direct injection of animal slurry into soil. Other technologies and systems to provide further solutions continue to be researched.

**Abstract:**

Animal-sourced foods are important for human nutrition and health, but they can have a negative impact on the environment. These impacts can result in land use tensions associated with population growth and the loss of native forests and wetlands during agricultural expansion. Increased greenhouse gas emissions, and high water use but poor water quality outcomes can also be associated. Life cycle analysis from cradle-to-distribution has shown that novel plant-based meat alternatives can have an environmental footprint lower than that of beef finished in feedlots, but higher than for beef raised on well-managed grazed pastures. However, several technologies and practices can be used to mitigate impacts. These include ensuring that grazing occurs when feed quality is high, the use of dietary additives, breeding of animals with higher growth rates and increased fecundity, rumen microbial manipulations through the use of vaccines, soil management to reduce nitrous oxide emission, management systems to improve carbon sequestration, improved nutrient use efficacy throughout the food chain, incorporating maize silage along with grasslands, use of cover crops, low-emission composting barns, covered manure storages, and direct injection of animal slurry into soil. The technologies and systems that help mitigate or actually provide solutions to the environmental impact are under constant refinement to enable ever-more efficient production systems to allow for the provision of animal-sourced foods to an ever-increasing population.

## 1. Introduction

Animal-sourced foods have been shown to be an important part of the human diet for both nutrition and health [1]. However, negative environmental impacts are among the most significant issues identified by the FAO resulting from producing animal-sourced foods [2]. They include land use change with its attendant consequences for deforestation and competition between the food (crop) and feed (forage), greenhouse gas emissions, and sustainable land and water use including the maintenance of soil carbon (C) stocks and the quality of freshwater. Unintended environmental impacts of pasture-based agricultural systems need to be managed and mitigated to achieve more acceptable and sustainable outcomes. The factors listed above are frequently interrelated, especially in developing countries where land use change drives substantial changes in the water, carbon, and nutrient cycles that affect soil, water, and atmospheric quality. However, even in developed countries, with relatively stable land use patterns, the impacts of pasture/grassland-based agriculture frequently exceed the limits imposed by society via government regulations and must be reined in to ensure the sustainable supply of food for local and global populations. The aim here is to undertake a systematic review of the current literature to examine these impacts that can result in land use tensions associated with population growth and the loss of native forests and wetlands during agricultural expansion, increased greenhouse gas emissions, and high water use with poor water quality outcomes. Technologies and practices that can be used to mitigate these impacts will be discussed.

These dynamic environmental factors are discussed in detail below, mostly in relation to pasture-based livestock production systems in developed countries and in keeping with a focus on grazing systems in countries such as New Zealand and Ireland. A key difference between pasture-based food production and cropping is the presence of the ruminant animal, and the unique ways in which the animal alters the water, C, and nutrient cycles compared with cropping systems, for example, in decoupling C and nitrogen (N) in the feed they consume. In doing so, they turn food that is not usable by humans (forages, crop residues, and agricultural byproducts) into high-value products and services [3]. Perceptions of animal welfare, whether positive or negative, are increasingly shaping consumer decisions regarding their dietary choices [4,5]. Pasture/grassland-based animal production systems are uniquely different from confinement systems with regard to, for example, the interactions between the animal and its immediate environment, and the types of diseases that affect their productivity and longevity. These issues are highly relevant to the challenge/opportunity for pasture-based food production to differentiate itself from other food or feed industries with respect to nature- and animal-positive attributes. Previously, the impacts of animal-sourced foods on human nutrition and health, herd management, animal health, and human–livestock relationships have been discussed [1]; here, the focus will be on the impact of food production from livestock on the environment and technologies used to mitigate consequences.

## 2. Land Use

Undeniably, cities are highly dependent on rural areas for the provision of food to ensure their survival. Before the global trade in food became commonplace, it was normal for food consumed by a city to be produced in the nearby countryside. Indeed, the first cities of major urban civilizations were often constructed close to fertile farming regions. Today however, with global trading, the countryside may end up feeding urbanized populations globally [6].

During the twentieth century, the global population increased by 230% and cultivated land area by 56% [7] largely due to a substantial decrease in forests, particularly tropical rainforests, resulting in a decline in biodiversity, increased soil erosion, and a significant change in the global carbon cycle [8,9]. These trends continued into the 21st century: Borrelli et al. [10] reported that from 2001 to 2012, the total global area in semi-natural vegetation (predominantly grassland, shrubland, and savannah) and cropland increased by 1.43 and 0.22 million km^2^ respectively, while the area under forest decreased by 1.65 million km^2^ (the net change is in balance but there were transitions among all three land uses classes, including crop to semi-native vegetation and vice versa).

Land use tensions driven by population growth, concomitant urbanization, and changing land use needs are occurring in both developed and developing countries [11]. For example, in northwestern Cameroon, competition is occurring between local crop farmers seeking a land use change from natural and traditional grazing areas into agricultural land [12]; in the Kilombero valley of Tanzania, pastoralists have been evicted to make way for an expanding sector of agricultural capital investments combined with a substantial increase in areas under environmental conservation [13]; in Ethiopia, rapid urbanization mainly induced by migration has been at the expense of agriculture, plantation, and mixed forests [14]; in Pakistan, cities have expanded, reducing productive agricultural land substantially [15]; a competition for land between family-based farming and state-supported agribusiness farming is occurring in Brazil [16]; the uncontrolled expansion of cities in southern Europe is onto agricultural land [17]; in Australia and New Zealand, tensions exist at the frontiers of urban and peri-urban, and also between the use of land for agriculture and forestry, primarily with exotic but also native species [18,19]. Residential and commercial development onto greenfield land around Auckland, New Zealand’s largest city, will inevitably lead to increased vegetable prices [11].

In a meta-analysis of 62 studies, predominantly from the tropics, “overlapping land rights, ethnic fragmentation, and corruption are the most frequently reported root causes [of conflict], followed by economic inequality, migration, and high dependence on agriculture” [20]. Expansion into subtropical and tropical forests to allow for the development for both cropping and animal-based agriculture [21,22,23], while causing major environmental damage and a reduction in biodiversity [24,25], can also result in wider social impacts for “forest-dependent” societies [26]. 

A meta-analysis [27] of ecological studies undertaken in Europe determining land use on biodiversity has concluded the following.

Traditional extensive grazing in southern Europe and mountainous ecosystems created a mosaic of habitats resulting in biodiversity;Short-term abandonment following overgrazing had positive effects in central Europe and lowlands but not in southern European mountains;The abandonment of long-term traditional grazing activities in mountainous ecosystems resulted in the extinction of populations of species tightly linked with open habitats; andIn lowlands, the abandonment of some grazing patches augments habitat diversification and creates new habitats for more species, but overgrazing caused a significant decrease in biodiversity.

A system that balances population growth and economic development with environmental imperatives has been labelled as “sustainable intensification” [28,29]. However, achieving this outcome will be context dependent, based on farm type and location, whether it is peri-urban agriculture, large farms situated on prime agricultural lands, or smallholder farms often associated with subsistence farming [30]. 

Crop and livestock farming, and the land use change that often accompanies it, such as clearing forests and drying out wetlands, has been estimated to account for more than a fifth of global carbon output [31]. Further forest and wetland conversion is being halted, at least in developed countries: the challenge of balancing environmental impacts and meeting the nutritional needs of a growing world population with currently farmed land is significant. Globally, the area of land not currently used for food (crop), feed (forage), or bioenergy production that could practically and legally be converted to those uses is extremely limited [32]. Proposed solutions to this situation include the conversion of degraded land in Latin America and sub-Saharan Africa to cropland, improved yields on existing land, and reducing current food waste. 

Comparisons between livestock and cropping systems for the production of human edible food tend to overlook (a) the amount of feed that animals consume that is not useable by humans [33], with 86% of the total feed consumed by animals coming from materials that are currently not eaten by humans, and (b) the impact of even modest improvements in feed use efficiency on land currently used for animal-sourced food production (land that is generally unsuitable for cropping) without the need for more deforestation to release land for cropping [3]. Despite these, there is also a need to work towards reducing the use of cereal grain for ruminant (and monogastric) livestock, while noting that yields for feed-grain tend to be higher than for human use, so a direct substitution is not possible. Bertsch [34] notes that “In developed countries, 56% of cereals produced are for livestock feed, and 23% in developing countries. Globally, 37% of cereal production goes to animal protein production”. On a global scale 40% of crop calories are used as livestock feed with a conversion ratio of about four kcal of crop product per one kcal of animal product [35]. However, there is considerable variation across regions and management systems with low crop calories used in production systems for ruminants based on fodder and forage, while large values are usually associated with production systems for non-ruminants (namely, poultry, and pigs) fed on crop products such as grain in direct competition with humans. 

Criticisms of livestock production have led to the call to phase out animal production [36] and to develop plant-based meat alternatives. The latter, depending on production practices, could have considerably lower environmental footprints in terms of the carbon footprint per unit of product [37,38,39]. A life cycle analysis from cradle-to-distribution, excluding the greenhouse gas emission from retail, restaurant, or at-home use, and end-of-life stages, has shown that novel plant-based meat alternatives can have an environmental footprint lower than that of beef finished in feedlots, but higher than for beef raised on well-managed grazed pastures (Table 1). The use of the metric CO_2_-eq/kg product or unit of protein/energy that is often used in life cycle analyses has been criticized as over-simplistic and not adequately representing the true impact and value of livestock products [40]. Their view is that “CO_2_-eq does not adequately reflect the different nature of CH_4_, the main GHG emitted from ruminant livestock systems, compared to (sic) CO_2_ and N_2_O in the atmosphere. On the other hand, kg product does not adequately consider the value of livestock: for example, nutritionally, they are generators of valuable co-products, whilst also being re-cyclers of byproducts, up-cyclers of nonproductive land, potential soil and biodiversity enhancers, and also offer social resilience platforms”. Global warming potential (GWP*), combining emissions (pulse) and changes in emissions levels (step) [41,42], is favored by many in animal production because it appears to result in a less punitive outcome for animals. However, as indicated by Manzano et al. [40], different metrics can be used in different scenarios to support different arguments. What is not in doubt is that achieving efficiency of production is of benefit to all parties.

Interestingly, the production of lab-grown meat can result in high levels of greenhouse gas emissions, although estimates are variable (Table 1). Non-renewable energy use was also very high for lab-grown meat at 290 to 373 MJ/kg, which was compared with 48 to 59 MJ/kg for dairy-based product and 27 to 37 MJ/kg for soybean meal-based product [43]. Energy use for feedlot beef production was 77 MJ/kg compared with the value for lab-grown meat of 108 KJ/kg [44]. However, another study by Tuomisto and Teixeira de Mattos [45] showed a much lower environmental impact (Table 1). More recently, Risner et al. [46] indicated that “the environmental impact of near-term animal cell-based meat (ACBM) production is likely to be orders of magnitude higher than median beef production if a highly refined growth medium is utilized for ACBM production”. The highly refined growth medium is required to reduce potential contamination. Another recent review of the environmental impact of lab-grown meat has concluded that “no complete nor consistent life-cycle assessment of cultured meat has been conducted, owing to the lack of information related to the processes and materials” [47].

**Table 1 animals-14-01353-t001:** Summary of the life cycle analysis (LCA) for meat from animals fed either through a feedlot or on grazed pasture, plant-based meat alternatives, and lab-grown meat measured as C footprint—kg CO_2_-eq per kg product. Further summaries of the life cycle analysis for both plant-based and lab-grown (cultured) meat are provided by UNEP [48] and Smetana et al. [49].

Production System				Comment	Reference
Meat from Feedlot-Fed Animals	Meat from Grazed-Pasture Animals	Plant-Based Meat	Lab-Grown Meat		
Beef +*22				Western Canada—lifetime GHG emissions	[50]
Dairy +3.8 to +6.2		Mycoprotein based +2.4 to +2.6			[51,52]
			+1.8 to +2.3		[45]
Beef +9 to +42; Dairy +1 to +2	Suckler herds +23 to +52 Extensive pastoral +12 to +129	+1 to +2		Review of numerous studies	[53]
+31			+7.5	Feedlot beef in upper midwest USA	[44]
Dairy +4.3 to +4.9 Chicken meat +5.2 to +5.8		Soymeal-based +2.6 to +2.8 Mycoprotein-based +5.5 to +6.1	+23.9 to +24.6	Cradle-to-plate life cycle	[43]
+150		+35			[54]
+33—US feedlot		+3.5 per kg beef		Total cradle-to-distribution impacts Beyond Burger and U.S. beef in feedlot	[55]
Feedlot from +6.09 to +6.12 due to soil erosion	Rotationally grazed systems moved from +9.62 to −6.65 due to soil erosion			Change due to inclusion of soil organic matter accumulation in analysis	[56]
+48.4				Full LCA for USA beef—includes feed production and feedlot	[57]
+33—US feedlot beef	−3.5 for grazed pastures			Rotational grazed beef can in some circumstances have a negative carbon impact due to soil C sequestration	[58]
+21.3				Full LCA for USA beef for feedlot and processing, etc.	[59]
Beef +48 to +210 Dairy +35 to +45 Sheep +80 to +190		+5 to +35	+15 to +40		[60]
+11 (feedlot finished beef)	−3.5 (grazed)	+3.5 (soy-based) +3 (pea-based)		Cradle-to-distribution LCA, but excludes GHGE potential of retail, restaurant, or at-home use, and end-of-life stages	[61]
	+42 to +235	+21 to +55		Range depends on functional unit and allocation method	[62]
	+6.01 for sheep and +8.97 for beef cattle			“Cradle-to-grave” for average NZ sheep and beef (weighted for traditional and dairy beef)—grazed pasture	[63]
			+4.9 to +25.2		[64]
	Traditional beef +10.09 Dairy beef +6.88 Sheep +6.01			NZ—Cradle-to-farm-gate GHG emissions per kg live-weight sold (kg CO_2_e kg^−1^ LW) for grazed pasture	[65]
+11 (dairy +40 (beef)			+2.2 to +24.8	Current benchmark; best and worst case for cultured meat	[66]

* The plus (+) sign indicates an overall net positiveincrease in kg CO_2_-eq per kg product.

A life cycle analysis of animal produce from New Zealand has shown that despite the long shipping distances involved, New Zealand beef, sheep meat [65], and dairy products [67,68] supplied to international markets have a full life-cycle carbon footprint at the lower end of other published estimates. However, the methods of calculation of these estimates do differ among authors across countries as noted by Mazzetto et al. [68]. The researchers found that “countries where milk is produced mainly as a pasture-based system had most of their footprint (>50%) associated with the emission of methane from enteric fermentation, whereas other countries (especially from Europe and North America) had a significant share of emissions from manure management, feed production, and fertilizer use”.

A study in Ireland using a life cycle assessment demonstrated that the substitution of synthetic nitrogen fertilizer with atmospheric N fixed by white clover has the potential to reduce the environmental impact of intensive pasture-based dairy systems in temperate regions [69]. The reduction was achieved through both the improvement in animal performance and the reduction in total emissions and pollutants. It is well understood that the improved pasture quality due to the inclusion of legume, such as white clover, is likely to result in higher milk and meat production per unit area than where legumes are absent and fertilizer N is used on grass-only pastures [70,71]. However, for the best results measured as maximum gross margins per ha and high levels of milk solids production per ha and per cow, a combination of N inputs from white clover and fertilizer, such that clover contents are 30–40% with N fertilizer rates of 100–200 kg N/ha/year, is required [72]. 

## 3. Greenhouse Gas Emissions

Despite O’Neil et al. [73] demonstrating that dairy cows grazing high-quality pasture have lower greenhouse gas emissions than those fed total mixed rations, it has been concluded by others that globally greenhouse gas emissions are lower for ruminants fed total mixed rations compared with those grazed on pasture [74] (Table 2). This is due largely to high emission intensities being associated with low productivity systems (Figure 1—[75]). This in turn is a result of the higher feed digestibility of total mixed ration diets compared with what is possible from grazed pastures, particularly the poor-quality forage in some developing countries [76,77]. The apparent conclusion is that greenhouse gas emissions are lower for animals fed high-quality feed (i.e., highly digestible, high protein, and high energy) than those fed low-quality feed. Views can be polarized with some concluding that very high quantities of beef consumption are climatically unsustainable, regardless of the CO_2_ equivalence metric [78], while others believe that the call to reduce the consumption of meat and other livestock products [79,80] is not fully supported by evidence (and should always indicate a starting point—much of the world population would benefit from an increased consumption of high-quality protein). While studies have frequently highlighted that beef production is responsible for intensive greenhouse gas emissions [81], relevant data supporting this position are not as widespread or as robust as they may first appear [82]. Emissions on pasture-fed systems are largely enteric methane production, which is shorter lived than long-lived emissions like nitrous oxide (N_2_O) and CO_2_, which are the main emissions associated with using total mixed rations based on the use of fertilizer to grow them, transportation, and land use changes associated with the production of feed crops [83,84]. 

Mitigation opportunities to reduce greenhouse gas emissions might include the following.

Feed type and quality: Improving the nutritive value of the grazed feed through replacing low-quality native pasture with improved higher-quality pasture increases the enteric methane emission (g/day) produced by ruminants but reduces the methane yield per unit of meat or wool produced [86,87]. The increase in dietary lipids that improves nutritive value through balancing the ratios of energy to protein in diets has been shown to reduce greenhouse gas emissions [88]. This has led to programs seeking to increase lipid levels in ryegrass [89,90]. It is also well known that the incorporation of species into pasture that express condensed tannins, which protect protein in the rumen will reduce greenhouse gas emissions [91,92]. Other forage species have been shown to reduce methane emissions when eaten such as biserrula (*Biserrula pelecinus*) [93,94], sulla (*Hedysarum coronarium*) [91,95], *Lotus corniculatus* [91,96], *L. pedunculatus* [97], and sainfoin (*Onobrychis viciifolia*) [98]. However, these species are agronomically inferior to the forages currently used and their management under grazing is a challenge. A program in white clover (*Trifolium repens*), the most used pasture legume in temperate areas, is set to achieve condensed tannin expression in leaf tissue through the use of a transcription factor taken from a closely related *Trifolium* species [99,100].Dietary additives such as oils, microalgae, macroalgae, nitrate, ionophores, protozoal control, phytochemicals from plant extracts, and 3-nitrooxypropanol have shown differing levels of efficacy in reducing methane production per kg dry matter consumed [101,102,103,104]. Macroalgae and 3-nitrooxypropanol have shown the greatest efficacy in reducing methane yield. The seaweeds *Asparagopsis taxiformis* and *A. armata,* when included at low concentrations in the feed of cattle and sheep, inhibit methanogenesis by up to 98% [105,106]. The active ingredient from these macroalgae are bromoforms (organic compounds that are classified as a probable human carcinogen by the US EPA, but can be found in chlorinated drinking water [107]). Bromoform inhibits an enzyme in the methanogenesis pathway [108]. Studies are mixed on whether there are negative impacts on animal health or food quality [109]. However, because bromoforms are rapidly metabolized by rumen microbes [110], to be effective they need to be included with feed at a rate of 0.4–1.0 mg/kg animal/day of bromoform [111]. For animals in pasture, the major difficulty is longevity of action. Further, to be cost effective the expense associated with wild harvest and indeed aquaculture production will need to be reduced [109]. Canola oil has been shown to reduce methane losses from cattle, but animal performance may be compromised due to lower feed intake and decreased fiber digestibility [112] Fumaric acid, which can utilize hydrogen (instead of it combining with carbon to form methane), has been disappointing as an additive [112,113,114], but when encapsulated in partially hydrogenated vegetable oil it suppressed methane formation by 19% [115]. The main ways that many of these additives reduce methane production is through reducing the number of rumen protozoa and inhibiting methanogen activity, increasing propionic acid production, which competes with methanogens for hydrogen, and inhibiting the activity of enzymes involved in methanogen activity [116].The breeding of animals with higher growth rates and increased fecundity [86,87,117]. Breeding ruminants with lower methane production has been shown to be a feasible option [118,119,120] with heritability of g methane/day of 0.29 ± 0.05, and for g methane/kg DMI of 0.13 ± 0.03 [121]. Breeding for animals with low methane production per unit of dry matter intake is unlikely to negatively affect fecal egg counts, adult ewe fertility, and litter survival traits, with no evidence for significant genetic correlations, but may reduce wool, live weight, and fat deposition traits [122].Rumen microbial manipulations through the use of vaccines [87,123,124,125]. A recent review has concluded that it is complicated to evaluate the real effectiveness of this strategy with few published studies that have directly assessed the complete approach from vaccination to enteric animal methane emission measurement [126]. Similarly, the antibiotic monensin as a rumen additive has shown some success in vitro but results from in vivo trials have been disappointing [113].Pasture management, which ensures grazing occurs when fiber content is low (e.g., prior to grasses maturing and flowering) has been proposed as a method of reducing methane emissions [91].Animal management that reduces age at first breeding [117] and age to slaughter [127] has been proposed as a means of reducing methane emissions. This is essentially a measure of efficiency that has underpinned the sheep and beef industry in New Zealand over the last 30 years [128].Soil management to reduce N_2_O emissions: Reduced tillage and use of a nitrification inhibitor when using N fertilizer on intensive pastures has been proposed as a means of reducing N_2_O emissions [129,130].

## 4. Water Use and Quality

Water is obviously crucial for growing the feed required for livestock production, whether that feed is grown on specialist cropping farms and transported to feed animals indoors or offered as pasture for animals to graze in situ. The US beef industry is predominantly based on the former, which [59] argues results in poor water use efficiency and landscape degradation. In an assessment of a wider range of beef production systems, Ridoutt et al. [131] concluded that “many low input, predominantly non-irrigated, pasture-based livestock production systems have little impact on freshwater resources from consumptive water use, and the livestock have a water footprint similar to many broad-acre cereals”. Improvements in livestock water productivity (protein produced per m^3^ of water) should be possible based on the wide range shown between livestock types, regions, and production systems [132]. However, Heinke et al. [132] also concluded that while opportunities to increase feed use efficiency (protein produced per kg of feed) exist for ruminants, the overall potential to increase their feed water productivity (feed produced per m^3^ of water) is low.

The in situ grazing systems used in countries like New Zealand and Ireland rely predominantly on natural rainfall to support pasture growth. Farms generally import only small amounts of water and nutrients in supplementary feeds. The over-use of water resources is not generally a concern, though ~25% of New Zealand’s milk production comes from irrigated pastures (mainly in the drier South Island of the country), and over-irrigation can markedly increase nutrient losses (especially N) to freshwater, as discussed further below. Irrigated pasture area in the South Island of New Zealand has doubled since the early 2000′s, but still comprises <5% of New Zealand’s total grassland area [133]. Although New Zealand overall (in total, not just agriculture) uses less than 5% of the available water [134], security of the future water supply under climate change and increased competition for water from other users, including the local and central governments as the guardians of sustainable environmental water flows for maintenance of freshwater quality and ecology [135,136], is a concern. An example of a major change in pastoral agriculture driven by water availability is the situation in southeast Australia, where irrigated dairy has virtually disappeared from northern Victoria/southern NSW Murray–Darling-fed regions because of reduced inflows and greater government environmental buybacks of water allocations [137]. 

The most significant water-related issue for the more intensive grazing industries of New Zealand and Ireland is nutrient losses to receiving freshwater bodies, especially rising loads of N and/or phosphorus (P) leading to increased incidences of algal blooms and higher rates of loss of key macroinvertebrate species [138,139]. An increased presence of pathogens from animal origins, such as *Escherichia coli*, and sediment run-off from steeper land, are also key concerns. All four contaminants have been highlighted recently [140]. Across NZ’s agricultural regions, there is a strong positive relationship between the extent of pastoral land use and the total N, P, and *E. coli* loads in local freshwater bodies (Table 3); the relationship is weaker for land used for cropping, and negative for areas under native forest. Simultaneously, public concern regarding the negative environmental impacts of “intensive” dairy farming has risen sharply, as reflected in calls in social media for (total) dairy cow numbers to be reduced [141]. In response, central and regional government environmental legislation since 2010 has progressively imposed limits on the nutrient discharges allowed from all primary production land uses but focuses especially on dairy and limits on N fertilizer use. 

Improving nutrient use efficiency through the food chain is a laudable goal. In the European Union, it has been estimated that only 18% of N used ends up in the eaten product with the remaining lost to the environment [143]. With 75 to 90% of consumed N in ruminant feed excreted in either urine or feces [144], the fate of this N in soil and subsequently in waterways is a reasonable concern. Nitrogen in ruminant excreta is not only a potential source of nitrate in waterways but is also a significant source of N_2_O [145]. The amount of N in urine is estimated to be about 40 to 50% of the total excreted [146,147]. Nitrogen excreted into dung or urine can have very different mineralization timeframes with N in dung slower to mineralize due to higher dry matter content [148]. Dietary factors such as condensed tannins and possibly phytochemicals that have the capacity to direct more N into dung away from urine could be beneficial [149,150]. 

Any form of intensive, extractive land use will have effects on the environment, including nutrient pollution [151]. This applies to all food production, not just grazing animals. An example of the general relationship between food production and environmental quality is shown in Figure 2. A key point is that the relationship is not linear: to the far right of the agricultural output axis in Figure 2, measures of environmental quality decline disproportionally to gains in production. This implies that a limited reduction in food output from near maximum levels (e.g., point C on the curve) towards some optimum (e.g., point D), would be beneficial. Nitrogen is especially difficult to contain within farm systems since its chemistry leads to multiple pathways of loss (in air as well as water) and soils have a limited capacity to store it. The more N that enters a farm system, the more N that is lost to the environment (in gaseous forms such as N_2_O, di-N, and ammonia, nitrate in drainage water, or in overland flow) [152]. 

The relationship shown in Figure 2 is useful in the sense that it directly compares two outputs—agricultural output (synonymous with food production) and environmental quality (synonymous with the output of, for example, nutrient or greenhouse gas emissions). Resolving the conflicts between meeting global food security while minimizing the environmental impacts of food production is best served by comparing outputs with outputs—rather than the common approach of relating inputs (usually of a single factor, e.g., fertilizer nutrient) to outputs. Although Figure 2 shows a simple, single relationship, in reality, there will be several relationships (lines) because the dynamics of different environmentally important factors are fundamentally different. For example, for N there is an asymptotic (saturating) relationship between the rate of N use and food production (as other factors come to limit crop or pasture/animal production) but an exponential relationship between the rate of N use and environmental quality (because once plant production is maximized, further N inputs will be lost to the environment). By contrast, relationships between the rate of energy supply for animals and food output can be linear if additional feed energy (dominantly carbohydrate/carbon) is imported to keep producing more meat or milk. Thus, different ways of producing food (such as dairy or meat), and different ways of increasing food production (such as using higher rates of N input or feed supplements), have very different effects on the relationship between food output and environmental impacts [154,155]. Re-casting analyses to compare outputs that account for these dynamics is fundamentally important for ensuring that advocacy for policies that are deemed to reduce impacts does not lead to solutions that have little impact, or even make the problems worse. 

One useful, and easily calculated, measure of the risk of N emissions to the environment is the farm-gate N surplus. This is the difference between the total amount of N imported to the system (mainly in fertilizer and feed, though biological N fixation must also be considered) and the amount of N exported from the system (in milk or meat, or in some cases in conserved feed that is sold off-farm) [156,157,158]. In the absence of an increasing accumulation of N in soil organic matter (which is rare in developed agricultural soils), the N surplus will inevitably find its way into the environment. Almost all dairy or meat farms will have an annual N surplus, therefore all farms will have a N footprint in the environment. The critical issue for managing that footprint is to maximize the efficiency with which imported N is converted to N in the product (i.e., food). In this context, the N use efficiency (NUE) is simply defined as the ratio of outputs to inputs expressed as a percentage. Of the two measures, the N surplus is more meaningful because it is in units of mass, and it is the mass of N that matters for the environment. N use efficiency is, however, a useful indicator to help manage the surplus, and can be improved by manipulating both outputs and inputs.

An analysis of physical data from 380 New Zealand dairy farms showed farm N surpluses ranging from 50 to 400 kg N/ha per year and NUE ranging mostly between 25% and 33% but with some markedly lower than 25% (solid lines in Figure 3). Similar numbers and intensification trends have been reported for pasture-based dairy farms in Australia [159]. Comparable experimental data from New Zealand farm systems studies reported by Macdonald et al. [160] showed generally higher amounts of N in product than the NZ farm data (higher milk production) and N surpluses in the range of 200 to 450 kg N/ha/year (open symbols in Figure 3). 

Figure 3 shows that a high NUE does not necessarily lead to a low N surplus and therefore, a low risk of N loss. Rather, NUE is an indicator of the potential for increasing the efficiency of N use through management, and this needs to occur in tandem with a reduction in N inputs to achieve financial and environmental sustainability. Simply reducing inputs without improving the efficiency of use of those inputs leads directly to lower milk production, as shown by tracing toward the origin on the NUE lines in Figure 3. A combination of lower inputs leading to a lower N surplus, and increased efficiency of use of imported N, is needed. Changing, for example, a supplementary feed with a moderate–high N content to one with low N content will result in only small reductions in environmental N losses if the system continues to operate with a high N surplus. Thus, both strategic changes to the farm system, e.g., adjusting the demand (mainly driven by the number of animals per hectare) to match the reduced feed supply, and tactical changes, e.g., adjusting the timing of fertilizer and feed inputs during the annual production cycle, are necessary. System-scale studies in NZ [163,164] and Ireland [165] have demonstrated approximately 30–50% reductions in nitrate leaching and/or nitrate concentrations in groundwater by applying integrated responses such as those listed above, with only small effects on the farm business profit. The farm dataset used in Figure 3 was characterized by a large variation among farms as noted also by de Klein et al. [157]. Hence, there is considerable potential for most farms to reduce their N surplus. Farm systems where N inputs total no more than 300 kg/ha per year from all sources (not just fertilizer, which should not exceed 150 kg N/ha per year; biological N fixation can contribute substantial amounts of N [166], and converting N and feed inputs to milk must be performed efficiently (NUE = 33% or higher)) are well-positioned to maintain high milk production with a relatively low N surplus and, therefore, relatively low N leaching risk. Such systems should also be highly profitable [160]. 

Similarly, analyses based on farm systems in NW Europe that incorporate some degree of direct grazing by dairy or beef animals illustrate that system changes such as incorporating maize silage along with grasslands, use of cover crops, low-emission composting barns, covered manure storages, and direct injection of animal slurry into soil “greatly reduce N losses” [167]. 

Solutions to the problems of declining freshwater quality, such as legislating for limited numbers of animals per hectare or banning artificial N fertilizer, need not be radical and reactive. There are proven, scalable, and adoptable solutions available to reduce the impacts and align farm emissions with the needs of a sustainable natural environment. In addition, there are technologies that can/could further reduce N losses (and may in some cases concomitantly reduce methane emissions), including the following.

N fertilizer used in conjunction with urease inhibitors such as N-(n-butyl)-thiophosphoric triamide and N-(n-propyl)-thiophosphoric triamide with an ability to reduce N_2_O and ammonia emissions while preserving yield [168]. However, caution has been called for from a meta-analysis that concluded that urease inhibitors applied with 20–30 kg N/ha per application in the spring and autumn are unlikely to increase plant dry matter yields and lead to improved NUE [169].Supplementary feed formulations including essential oils [170].The use of nitrification inhibitors such as dicyandiamide (DCD) or 4-methylpyrazole (4MP) have been shown to reduce environmental N emissions from urine patches in pasture systems [171,172], but DCD has been withdrawn from use because of small amounts of DCD residue found in NZ milk products [173].Plant breeding [174] to exploit genetic variation among and within species in traits that have the potential to improve NUE (such as condensed tannin content as discussed above [100]), internal and external critical N concentrations [175], protein degradability [176,177], and biological nitrification inhibition [178].Animal breeding to select for animals with lower methane per unit of dry matter intake has been successfully achieved but for sheep impacts on wool, live weight, and fat deposition, traits may be affected and need to be monitored [120,122].Combining traits in complementary forage species mixtures [179] rather than monoculture grass or simple two-species mixtures could substantially reduce N leakage to the environment. For example, in New Zealand, a combination of a N-fixing legume (e.g., white clover) with a N-demanding grass (e.g., perennial ryegrass, which has a relatively high critical internal N content) and a herb that inhibits nitrification in the soil and/or dilutes the N concentration of urine (e.g., plantain, [178,180,181]), has been shown to reduce N leaching by up to 80% in lysimeter studies [182] and 40% in field studies [183]. Proof of practice for this approach is currently underway in whole-farm systems experiments over multiple years [184].

## 5. Carbon Sequestration

Some of the differences between life cycle analysis comparisons and beef raised on grazed pastures could be due to the soil-sequestered carbon that might be possible from rotationally grazed pasture systems (summarized by van Vliet et al. [61] but taken from [55,56,57,58,59,185]) (Table 1). The significance of soil carbon sequestration aligns with other analyses for North American grazing systems when compared with feedlot systems [186]. Indeed, it has been estimated that since tillage-based farming began, those soils have lost 30% to 75% of their soil organic carbon [187]. Ruminant livestock are acknowledged as important for achieving sustainable agriculture where appropriate grazing management “can increase carbon sequestered in the soil to more than offset their GHG emissions and can support and improve other essential ecosystem services for local populations” [186]. Teague [186] proposed an Adaptive Multi Paddock (AMP) grazing system of “adjusting animal numbers to match available forage, using short grazing periods, leaving sufficient post-herbivory plant residue for regrowth, and providing long recovery periods to adaptively accommodate intra- and inter-seasonal variation in herbaceous plant growth”. However, in a review comparing grazing systems, it was concluded that “the vast majority of experimental evidence does not support claims of enhanced ecological benefits in intensive rotational grazing compared to (sic) other grazing strategies, including the capacity to increase storage of soil organic carbon” [188].

A further consideration is to determine the potential for further soil carbon sequestration in grazing systems where soil carbon is already high and close to saturation. This situation can occur in New Zealand where soils under grazed pasture, which were originally forested, have over time accumulated moderately high concentrations of soil carbon of 3.5% (to 300 mm depth) [189], noting that soil carbon accumulation occurs up to an upper limit or “C saturation level”, which is determined by a number of chemicals and biological mechanisms [190]. New Zealand soils contain on average 100 tonnes of organic carbon per hectare to a depth of 300 mm [191], which has been interpreted as being close to their effective stabilization capacity; further increases in soil carbon could be difficult to achieve [192]. 

It has been proposed that increasing soil carbon by 0.4% per year globally could compensate for the global emissions of greenhouse gases by anthropogenic sources [193], although it is a contentious issue [194,195]. Minasny et al. [193] concluded that “as a strategy for climate change mitigation, soil carbon sequestration buys time over the next ten to twenty years while other effective sequestration and low carbon technologies become viable”. However, the opportunity to increase soil carbon depends on land use and management [196]—that management being past, present, and future.

Management decisions that might maintain or increase soil carbon are complex and often the best answer is “it depends”. The following factors have been shown in some instances to impact soil carbon.

Fertilizer application: N inputs (10 to 20 kg N/ha/year) to low-fertility grasslands can increase soil carbon [197]. However, whether or not N inputs are associated with increased soil carbon depends on grazing intensity [198,199]. A process-based model of the dynamics of carbon and N cycling between plants, soils, and animals in grazed temperate pastures indicated that the optimal N input for balancing food production, carbon sequestration, N loss to the environment, and greenhouse gas emissions in New Zealand is approximately 150 kg/ha N fertilizer [154]. Alternatively, P fertilizer application appears to have little effect on soil carbon accumulation following conversion from native vegetation to grasslands for grazing [196,200].Irrigation can result in variable and contradictory impacts on soil carbon. In desert and semi-arid areas, irrigation can increase soil carbon substantially, while in humid environments, no consistent effects have been observed [201]. Whitehead et al. [192] concluded that “no change or decreases in soil carbon stocks in response to irrigation in humid climates but increases could be expected at more arid sites where plant productivity is very low prior to irrigation”. In New Zealand, irrigation has been shown to decrease soil C [196,202] due possibly to effects on soil N levels in different soil types and management systems [203], whereas in arid and semi-arid environments, irrigation might be expected to increase soil C stocks due to increased plant growth and inputs to soils [201]. However, under irrigation it is likely that N_2_O emissions will increase by up to 140% [201].Refraining from draining peaty soils, which contain high quantities of soil C (and which can lose soil organic matter through the oxidation of organic matter after drainage) [204].Use of supplementary feed, such as hay or silage, which may result in small increases in soil carbon on paddocks where it is used but may also result in a small decrease in soil carbon at locations where it is produced due to the “length of time between harvest and re-establishment of the new crop, maximizing returns of organic residues, and adopting minimum tillage and direct-drill methods to reduce disturbance” [192].Application of manure and dairy effluent can increase soil carbon [205], although “the percentage of carbon retained in the soil is low, at approximately 4% of the total carbon applied” [192].Resowing of pasture can lead to a net loss of soil carbon and can be minimized by ensuring minimal soil disturbance and reducing time when the soil surface is left bare [192,206].Increased forage production also tends to increase soil carbon [207]. The Conant et al. review [207] indicated that the main drivers were use of more permanent pasture, improved grazing management, use of legumes, and increasing earthworm numbers. Grazing intensity can also impact soil carbon loss or accumulation. Overgrazing is generally considered to result in reduced soil carbon [197,198,208,209]. However, some studies have shown decreased soil carbon at both high and low grazing frequencies but most often the maximum accumulation occurred at a moderate grazing intensity [210,211]. This depends on balance—Parsons et al. [198]. A comparison of C4 and C3 gases in a meta-analysis has shown that higher grazing intensity results in increases in soil carbon in C4 grasslands but decreases in C3 grasslands [212]. This difference could be due to the high lignin levels in C4 grasses, which slow their decomposition and subsequent carbon release [213]. However, it is generally accepted that soil organic matter is greater in grazed pastures than non-grazed grasslands or land used for row crops or hay production [214].Species and diversity of species used: Whitehead et al. [192] concluded that forage species with deeper rooting and higher fine-root density at greater depths could increase soil carbon stocks. However, evidence that increased species diversity would increase soil carbon is inconclusive. The Jena Experiment setup in 2002 in Germany to investigate the effects of plant diversity on element cycling and trophic interactions do support increased soil carbon with increased pasture diversity, but the research involved mowing 2–4 times per year, i.e., there was no food production aspect [215].Use of biochar as a soil amendment may lead to an increase in soil carbon levels but its use as a widespread amendment to pasture soils is in many cases impractical [192].Full inversion tillage, which seeks to bury topsoil with high carbon levels to depths below 40 cm while bringing to the surface soil with a high carbon saturation deficit [192]. This would be achievable only on flat to moderately contoured sites and would be useful only where the soil carbon value for the topsoil is at least twofold greater than that of the subsoil [216].

Introduction of deep burrowing earthworms and dung beetles: Earthworms have been shown to transfer carbon in dung from the surface to depths of up to 30 cm [217] and stimulate carbon stabilization. However, other studies have shown no effect of earthworms on soil carbon content but rather a net increase in soil greenhouse gas emissions [218]. Tunnelling dung beetles are known to bury dung at depth in the soil [219] but efficacy of increasing soil carbon can be variable in New Zealand pastures [220]. 

A list of criteria that are appropriate to assess the success of changes in farm management practices to maintain or increase soil carbon stocks is provided by Whitehead et al. [192]. The authors remind us that “increase in on-site carbon sequestration must be greater than the emissions of all greenhouse gases associated with life cycle analysis of establishing and maintaining changes to all farming operations”.

## 6. Concluding Comments and Looking to the Future

It is acknowledged that societal demands for improved environmental integrity and sustainability mean that issues associated with animal-sourced foods that might negatively affect the environment need to be actively researched and managed. Impacts needing attention primarily include air, water and land quality, and animal health and welfare. Options for mitigating greenhouse gas and N_2_O emissions from systems using grazed pasture include the following.

Increased use of white clover and plantain in pasture seed mixtures;Ensuring that pasture is composed of forage species that are highly digestible with high protein and high energy, and low fiber content;Using ruminant animals with higher growth rates and increased fecundity;Reduced tillage when resowing and use of a nitrification inhibitor when using N fertilizer on intensive pastures;Including forages with levels of condensed tannins and possibly other phytochemicals that reduce methane emissions and do not affect palatability;Matching the use of fertilizer N with the demand driven by the number of animals per hectare to ensure the efficient conversion of N and feed inputs to milk and meat;Direct injection of animal slurry into soil where this is feasible;Maximizing carbon sequestration in soil where and when this is possible through reducing soil disturbance and fallowing; andAvoiding overgrazing, which can negatively affect persistence and result in soil disturbance through resowing.

In systems where supplementation can be introduced during feeding sessions, then use of dietary additives such as oils, microalgae, macroalgae, nitrate, ionophores, protozoal control, phytochemicals from plant extracts, and 3-nitrooxypropanol may reduce methane production. The use of vaccines is also a proposed option.

Animal-sourced foods are an important part of the human diet and while some unintended consequences associated with the environment have occurred, technologies and systems to provide solutions to these are available and under refinement.

## Figures and Tables

**Figure 1 animals-14-01353-f001:**
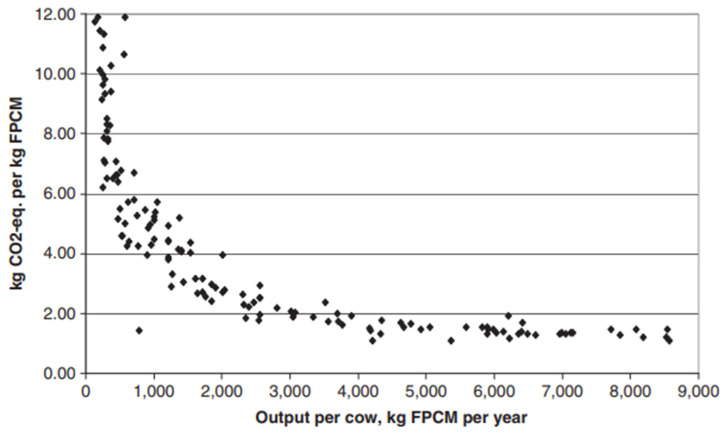
The relationship between total greenhouse gas intensity and output per cow. Each dot represents a separate country. FPCM—fat- and protein-corrected milk. While individual countries were not identified, milk yield per cow was below 1000 kg/cow/year for countries within sub-Saharan Africa and south and southeast Asia, and above 3000 kg/cow/year for countries within North America, western Europe, eastern Europe, and Oceania. Taken from Gerber et al. [75]. Copyright clearance under license number 5741800369452 provided by Elsevier.

**Figure 2 animals-14-01353-f002:**
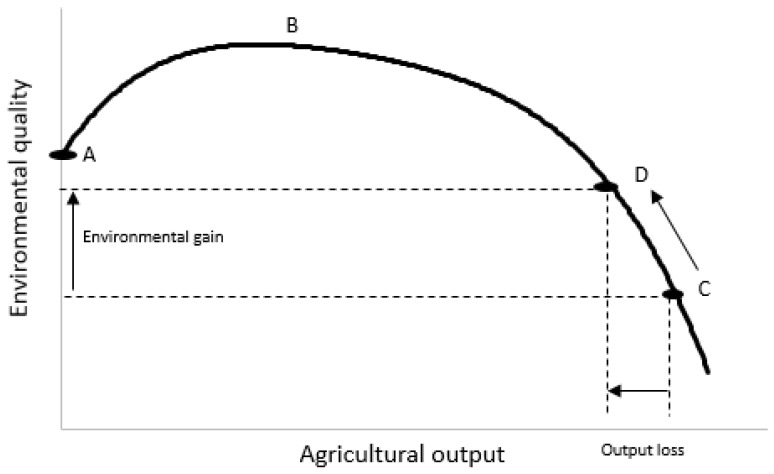
General relationship describing conflicts between product output (*x*-axis) and environmental outcomes (*y*-axis) in agriculture. Point A equates to no agricultural activity; between points A and B, environmental and agricultural outcomes both improve; C equates to a point where all incentives favour agricultural output and activity intensifies without counter-balancing environmental incentives; point D is reached when societal demands for environmental protection outweigh relatively small gains in further agricultural intensification. Taken from McInerny [153]. Reproduced with permission published by Cambridge University Press (License number 5778780809075).

**Figure 3 animals-14-01353-f003:**
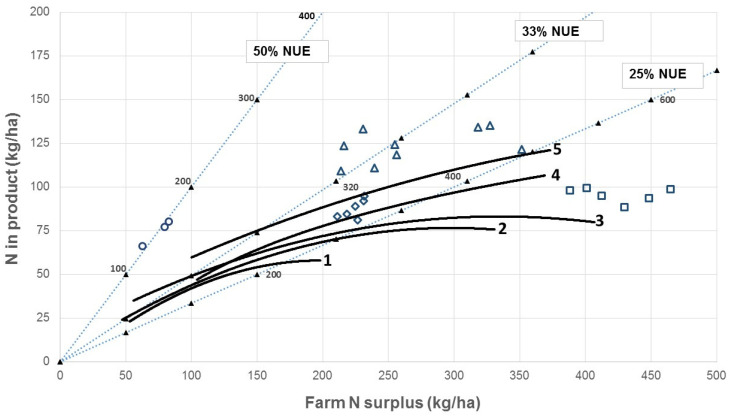
Relationships between N surplus and N output in product in grazed dairy systems. Diagonal lines join points of equal N use efficiency (NUE; 50%, 33%, and 25% are shown for comparison). Solid symbols and numbers on these lines indicate total N input (kg N/ha per year). Solid lines are fitted curves for commercial farms operating according to the definitions of systems 1, 2, 3, 4, or 5 in the New Zealand dairy industry, in increasing order of intensity of production (after Hedley et al. [161]). Open symbols are means for each of three years for treatments in a farm system experiment: circles = no N fertilizer or imported supplement; diamonds = 200 kg N fertilizer/ha/y, no supplements (2 stocking rates); squares = 400 kg N fertilizer/ha/y, no supplements (2 stocking rates); triangles = 200 kg N fertilizer/ha/y with supplements (three different types) [160]. Taken from Chapman et al. [162]. Copyright owned by D.F. Chapman (co-author).

**Table 2 animals-14-01353-t002:** Emissions and emission intensity associated with the global production of milk and meat by ruminant species and feed type. Adapted from Gerber et al. [74], using GLEAM, the Global Livestock Environmental Assessment Model.

Species	Feed System	Emissions (Million Tonnes CO_2_-eq)		Emission Intensity (Kg CO_2_-eq/kg Product)	
		Milk	Meat	Milk	Meat
**Dairy**	Grazing ^1^	227	104	2.9	21.9
	Mixed ^2^	1104	382	2.6	17.4
	**Total**	**1331**	**486**	**2.6**	**18.2**
**Beef**	Grazing ^1^	-	875	-	102
	Mixed ^2^	-	1463	-	56
	**Total**	**-**	**2338**	**-**	**67**
**Sheep**	Grazing^1^	30	76	9.8	23.8
	Mixed ^2^	37	115	7.5	23.2
	**Total**	**67**	**191**	**8.4**	**23.4**
**Goat**	Grazing ^1^	18	27	6.1	24.2
	Mixed ^2^	44	84	4.9	23.1
	**Total**	**62**	**111**	**5.2**	**23.3**
**Total from grazing systems ^1^**		**275**	**1082**	**6.3**	**43.0**
**Total from mixed rations ^2^**		**1185**	**2044**	**5.0**	**30.0**
**Grand total**		**1460**	**3126**	**5.4**	**33.0**

^1^ Grazing production systems are defined as livestock production systems in which more than 10% of the dry matter fed to animals is farm-produced and in which annual average stocking rates are less than ten livestock units per hectare (ha) of agricultural land [85]. ^2^ Mixed production systems are defined as livestock production systems in which more than 10% of the dry matter fed to livestock comes from crop byproducts and/or stubble or more than 10% of the value of production comes from non-livestock farming activities [85].

**Table 3 animals-14-01353-t003:** Correlation (r values) between water quality variables and land use in New Zealand. From Davies-Colley [142].

Variable	Land Use		
	Pastoral	Cropping and Horticulture	Native Forest
Total N	+0.85	+0.45	−0.39
Total P	+0.70	+0.24	−0.32
Visual clarity	−0.45	−0.24	+0.30
*Escherichia coli* presence	+0.80	(0.17)	−0.34
NZ land area (km^3^)	107,672	4174	65,675
Total NZ land area (%)	39.6	1.5	24.1

## Data Availability

Not applicable.

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
