# Peer review of "Improving Human Diets and Welfare through Using Herbivore-Based Foods: 2. Environmental Consequences and Mitigations"

_animals, 2024, doi:10.3390/ani14091353_

Round 1

Reviewer 1 Report

Comments and Suggestions for Authors

In the manuscript, the authors examine Improving human diets and welfare through using herbivore-based foods (Environmental consequences and mitigations).

The purpose and objectives of the study were completed in full.

The review presents the results of a thorough study on the problem of the impact of agriculture on the environment.

The authors do not allow incorrect citation of literary sources.

However, the following observations can be made.

1. Introduction. There is no research purpose (at the end of the section).

2. Page 2, last paragraph. I propose to highlight the points of the research methodology. This structure will be more understandable! It is necessary to indicate from which database the data was obtained and what period of time it concerns!

Comments on the Quality of English Language

The text of the manuscript is written in good English.

However, there are some typos in the text.

I recommend minor editing of the English language of the manuscript.

Reviewer 2 Report

Comments and Suggestions for Authors

This is a very well researched and written summary of the impact of animal-based production, specifically in relation to grazing, on the environment. 

Overall the paper is very well written however, there are some minor inconsistencies with the use of abbreviations and some terminology that needs to be addressed. The main example of this is the interchangeable use of nitrogen or N and carbon or C throughout the paper. Please either use nitrogen or N consistently throughout the paper. The same goes for carbon or C. I have highlighted some but not all of the instances where this should be corrected. Carbon Dioxide vs CO2 is another one where both terms are used interchangeably.

Another minor issue is, there seems to be some inconsistency with the use of double spaces to start sentences in some parts of the paper and single spaces in other parts. Again, I have highlighted some but not all of these instances. 

Section on water quality - could use more detail on possible solutions to mitigate nutrient loss impacts on water quality. Fro the climate change and carbon sequestration sections a list of bullet point solutions is provided. Should the same be provided for this section?

The tables also do not appear to be in the correct format, or have same size font etc. as the rest of the document. This is probably just a formatting issue but it needs to be rectified.

Specific comments are included in the annotated PDF attached. 

Reviewer 3 Report

Comments and Suggestions for Authors

The article concerns an interesting and important issue: the impact of food production from farm animals on the environment and technologies used. The article is a review of research and the modern literature. The summary of life cycle analysis (LCA) for meat from animals fed either through a feedlot or on grazed pasture, plant-based meat alternatives and lab-grown meat measured as C footprint - kg CO2-eq per kg product is very interesting. The explanation of the "+" sign is missing under table 1. The article also presents balanced arguments regarding plant-based meat substitutes and meat grown in the laboratory. It is important to present the emissions and emission intensity related to global milk and meat production by ruminant species and feed type.

Could it be possible to indicate in Figure 1 which specific countries have high emission intensity and low productivity (or vice versa)?

Is there no error in repeating "outputs" in the sentence "Re-casting anal-yses to compare outputs with outputs and accounting for these dynamics is fundamentally important.." (p.13)?

The paper also discusses the problem of carbon sequestration and water use efficiency associated with animal production.

The work also contains practical tips on what to do in animal production to reduce greenhouse gas emissions, nitrous oxide and methane from systems using pastures.

The conclusions are consistent with the argumentation and research literature presented.

The bibliography is appropriate, sufficient and includes many contemporary scientific articles.

Best regards
